# Itaconic Acid Increases the Efficacy of Tobramycin against *Pseudomonas aeruginosa* Biofilms

**DOI:** 10.3390/pharmaceutics12080691

**Published:** 2020-07-22

**Authors:** Duy-Khiet Ho, Chiara De Rossi, Brigitta Loretz, Xabier Murgia, Claus-Michael Lehr

**Affiliations:** 1HIPS–Helmholtz Institute for Pharmaceutical Research Saarland, HZI—Helmholtz Center for Infection Research, D-66123 Saarbrücken, Germany; chiaraderossi@gmx.de (C.D.R.); brigitta.loretz@helmholtz-hips.de (B.L.); 2Department of Pharmacy, Saarland University, D-66123 Saarbrücken, Germany

**Keywords:** tobramycin, ciprofloxacin, itaconic acid, itaconate, *Pseudomonas aeruginosa*, biofilms, infections

## Abstract

The search for novel therapeutics against pulmonary infections, in particular *Pseudomonas aeruginosa* (PA) biofilm infections, has been intense to deal with the emergent rise of antimicrobial resistance. Despite the numerous achievements in drug discovery and delivery strategies, only a limited number of therapeutics reach the clinic. To allow a timely preclinical development, a formulation should be highly effective, safe, and most importantly facile to produce. Thus, a simple combination of known actives that enhances the therapeutic efficacy would be a preferential choice compared to advanced drug delivery systems. In this study, we propose a novel combination of an anti-inflammatory agent—itaconic acid (itaconate, IA)—and an approved antibiotic—tobramycin (Tob) or ciprofloxacin (Cipro). The combination of Tob and IA at a molar ratio of 1:5 increased the biofilm eradicating efficacy in the strain PA14 wild type (wt) by ~4-fold compared to Tob alone. In contrast, such effect was not observed for the combination of IA with Cipro. Subsequent studies on the influence of IA on bacterial growth, pyocyanin production, and Tob biofilm penetration indicated that complexation with IA enhanced the transport of Tob through the biofilm. We recommend the simple and effective combination of Tob:IA for further testing in advanced preclinical models of PA biofilm infections.

## 1. Introduction

The increasingly urgent challenges posed by antimicrobial resistance require new therapeutic options complementary to established antibiotic therapies [1]. Especially considering extracellular bacterial infections, the formation of sessile biofilms is one of the crucial survival and defense strategies of pathogens leading to persistent infections [2,3]. Biofilms are surface-bound bacterial communities embedded in a self-generated extracellular matrix (ECM) composed of mainly polysaccharides, proteins and extracellular DNA (eDNA) [2,4]. Although the development of antimicrobial resistance is complex and involves multiple factors [2,5], the biofilm matrix, which is a biological barrier to penetration of antimicrobial agents, might be considered as the initial physical and biochemical cause of resistance to antibiotics [1,3,4]. 

One particularly virulent example is the persistent pulmonary infection caused by *Pseudomonas aeruginosa* (PA) in cystic fibrosis (CF) patients. Once PA invades the CF lung, its recurrence rate is almost a hundred percent, even after being treated with the most aggressive antibiotic therapy [6]. Clinical records have continuously revealed that inhaled antibiotics are the most effective approaches to maximize the therapeutic effects, recover pulmonary function, and most importantly, improve the quality of life of patients [1]. Despite intensive focus on the search for new anti-infectives, there is only a limited number of clinically approved inhalation therapeutics. Especially for pulmonary PA infections associated in CF, there have been only four inhaled antibiotics namely, tobramycin, colistin, levofloxacin and aztreonam being approved in Europe, while inhaled ciprofloxacin formulation has been studied in clinical trials. These antibiotics, however, cannot entirely eradicate bacteria embedded in biofilms and mucus [1,7]. To combat this kind of bacterial resistance, most recent research has mainly aimed to propose more potent agents [8], combinations of agents [6,8], or better delivery methods [9,10,11,12], which could have a prolonged and high enough bioavailability at the site of action and especially inside the biofilms by crossing and/or weakening this biological barrier. Notably, the co-administration of an antibiotic and a pathoblocker (e.g., *quorum* sensing inhibitor (QSI) which is not supposed to kill bacteria but affect biofilm formation) has shown advantages over antibiotic monotherapy and is a promising strategy to overcome the growing resistance problem in pulmonary PA infections [11,13,14,15] Biofilm penetrating nanoparticles based on self-assembled squalenyl hydrogen sulfate enabled the simultaneous co-delivery of hydrophobic QSI and hydrophilic tobramycin (Tob), leading to a significantly enhanced (~16-fold) efficacy of Tob to eradicate PA biofilms [11]. Although this formulation was proposed for use as inhalation therapy, like many other advanced drug delivery systems, it requires further optimization and must undergo full preclinical development. As a result, there is just a limited number of inhalable antibiotic nanopharmaceuticals reaching clinical trials, which are based on more established technologies and approved excipients, e.g., liposomes [16,17] or solid lipid formulations [1,6,18].

Alternatively, the combination of approved antibiotics with drugs that would enhance biofilm accessibility, hit different targets and benefit the recovery of lung function represents a more straightforward approach that may theoretically overcome the limitations of the treatment with a single antibiotic [8]. A successful example is the combination of the two antimicrobials, fosfomycin and Tob (ratio 4:1 *w/w*), as aerosolizable solution [19]. The formulation with the negatively-charged fosfomycin enhanced the active penetration of Tob in PA biofilms, which resulted in greater biofilm eradicating efficiency and most importantly lowered the resistance frequency to Tob [20]. Such formulation was reported in phase II in 2012 [21] and considered for phase III clinical trials [22]. The later studies on the combination of fosfomycin and other aminoglycosides, colistin or ciprofloxacin (Cipro) also showed enhanced PA killing efficacy [23]. However, all these antimicrobials aim to interfere with bacterial growth, which intrinsically puts stress on bacteria [24], and therefore the aforementioned combinations did not suppress the emergence of fosfomycin resistance as reported in prior works [23].

Itaconic acid (itaconate, IA), is a common industrial precursor for polymer synthesis [25], which has been surprisingly identified as an important metabolite induced during activation of immune cells. IA is produced in the metabolism of mammalian macrophages and modulates inflammation, e.g., inhibiting macrophage production of pro-inflammatory cytokines IL-6, IL-12, IL-1β as well as *NO* and *ROS* [26]. This simple molecule could become increasingly important to medicine, biology, and biotechnology [27]. IA contributes to an efficient immune response, especially against intracellular pathogens [28,29]. Furthermore, IA biological importance has also been proved by its contribution in loss-of-function mutations in *ACOD1*—the enzyme that catalyzes IA synthesis—which are extremely rare in human [27]. In contrast, extracellular pathogens, like PA, are able to encode enzymes that degrade IA to overcome the host defenses [30,31]. Biomedical uses of IA are in discovery and development phases. Scientists, e.g., Mills et al. [32], have attempted to increase the IA intracellular concentration to enhance the effects in the immune response. To our knowledge, the administration of IA against extracellular infections, in particular PA biofilms, has not been reported. In this work, we hypothesized that a combined formulation that locally increases both IA and antibiotic concentration in the lungs would be an interesting approach to treat PA infections in the context of CF. Therefore, we investigated for the first-time the antimicrobial properties and biofilm eradicating efficacy of the combination of IA with either Tob or Cipro, the *quorum* sensing inhibitory effects of IA, and the influence of the combination with IA on Tob biofilm penetration.

## 2. Materials and Methods 

### 2.1. Chemicals and Bacteria

Tobramycin (Tob, molecular weight (Mw) 467.5 Da), ciprofloxacin (Cipro, Mw 331.3 Da), itaconic acid (IA, Mw 130.1 Da), peptone, Tween20 and casein were obtained from Merck. Salts and organic compounds of analytical grade (NH_4_Cl, KCl, tris-HCl, chloroform, glycerol, glucose, formic acid and MgSO_4_·7H_2_O) were obtained from Merck and VWR. Yeast extract was obtained from Fluka. Bacto™ Tryptone were obtained from BD Biosciences. Luria Bertani (LB) agar was obtained from Carl Roth. Gibco^®^ PBS was obtained from Life Technologies. Purified water was produced by Milli-Q water purification system (Merk Millipore, Billerica, MA, USA)

The *Pseudomonas aeruginosa* strain PA14 wild type (wt) was obtained from ATCC or DSMZ, the German Collection of Microorganism and Cell Cultures GmbH (PA14 = DSMZ19882), stored in glycerol stocks at −80 °C. Bacteria were cultured in a minimal proteose peptone glucose ammonium salt (PPGAS) medium. PPGAS was composed of 1 g/L NH_4_Cl, 1.5 g/L KCl, 19 g/L Tris-HCl, 10 g/L peptone, 5 g/L glucose and 0.1 g/L MgSO4·7H_2_O. The medium was adjusted to pH 7.2 ± 0.2 and sterilized before use. Agar solution was composed of 15.5 g/L LB agar, 10 g/L peptone, 5 g/L NaCl and 5 g/L yeast extract, the solution was sterilized before being plated.

### 2.2. Test Samples Preparation

The stocks of Tob, Cipro or IA were prepared in phosphate-buffered saline (PBS) prior to assays, which served as controls. Tob combined with IA (Tob:IA) and Cipro combined with IA (Cipro:IA) solutions (molar ratio [antibiotic]:[IA] of 1:5) were prepared using the same procedure, as follows: A highly concentrated combined solution was prepared in MilliQ water to allow saturated complexation of the antibiotics and IA molecules. The solution was then diluted in PBS prior to assays. The molar ratio of [Cipro]:[IA] was designed at 1:5 to have relatively equivalent IA dose as used in combination with Tob, although there is only one amine group in Cipro.

### 2.3. Minimum Inhibitory Concentration (MIC) Assay

The antimicrobial activities of the single compounds, including Tob, Cipro and IA, as well as the combinations of Tob:IA and Cipro:IA (molar [antibiotic]:[IA] of 1:5) were investigated by standard microbroth dilution assays with planktonic PA14 wt in 96 well-plates. A suspension of PA14 wt prepared from mid-log cultures in PPGAS medium was first centrifuged and diluted to OD600 (absorption at 600 nm) 0.02, which corresponds to approximately 2 × 10^7^ cfu/mL (cfu, colony-forming units). Bacteria-containing wells were then treated with the respective samples. Concentration was serially diluted over a range of 0.315–300 μg/mL. Bacteria incubated with PBS served as control. After incubation for 16h at 37 °C, inhibitory concentration (IC) IC90 values were determined by sigmoidal curve fitting of absorption values (600 nm) that were measured on a Tecan microplate reader. The IC90 values are defined as concentrations at which the growth of bacteria is inhibited by 90%. Three independent experiments were conducted in triplicate.

### 2.4. Pyocyanin Assay

Pyocyanin assay was carried out as reported in prior studies [11,33,34]. This assay is to evaluate the inhibitory effects of a compound on pyocyanin production level of planktonic PA14 wt. As such, the tested compound is not supposed to kill or inhibit planktonic bacterial growth. Thus, antibiotics and their combination with IA were not included in this assay. Only IA was investigated in order to explore its potential *quorum* sensing inhibitory effects. Briefly, a single colony of PA14 wt was picked from an agar plate after 16 h growth at 37 °C, and cultured in 10 mL PPGAS. Following 16 h of aerobic growth (shaking at 200 rpm and 37 °C), cultures were centrifuged at 7.450× *g*, washed twice with fresh PPGAS, and finally diluted to a final OD600 of 0.02 in fresh PPGAS, which corresponds to approximately 2 × 10^7^ cfu/mL, and distributed into 24 well-plates, 1.5 mL each well. IA was added to bacteria-containing wells (1:100 dilutions, final PBS addition of 1% *v/v*) at concentration of 300 µg/mL, which is the highest IA concentration used in the further MBEC assays. PA14 wt treated with PBS served as controls. After further incubation for 16 h under aerobic conditions, pyocyanin was extracted from the bacterial culture with chloroform and re-extracted with 0.2 M HCl solution. Pyocyanin levels were determined by the absorption at 520 nm of the extracted aqueous solution, which was normalized to the cell growth measured as OD600. A 0.2 M HCl solution served as blank samples. Three independent experiments were conducted in triplicate.

### 2.5. Minimum Biofilm Eradicating Concentration (MBEC) Assay

MBEC assay was performed as reported in prior studies [11,35]. PA14 wt was grown and prepared in fresh PPGAS before being diluted to a final OD600 of 3.0, which corresponds to approximately 10^15^ cfu/mL, and distributed into 96 well-plates, 0.2 mL in each well. The 96 well-plates were then kept under aerobic conditions (37 °C, without shaking) for 24 h to allow biofilm formation. Afterwards, the planktonic bacteria were removed, and the 24 h-old biofilms were washed twice with PBS and fed with fresh PPGAS medium. Stock solutions of single drug Tob and Cipro, as well as the combination of Tob:IA and Cipro:IA (molar [antibiotic]:[IA] of 1:5) in PBS were diluted (1:100 dilution) and added to PA14 bacterial biofilm-containing wells to obtain final concentrations of Tob or Cirpo ranging between 6.25–200 µg/mL. Biofilms treated with PBS and 300 µg/mL IA only served as controls and were to exclude that IA could have an effect rather on some virulence, survival mechanism in biofilm than on direct bactericidal function. After further incubation for 24 h at 37 °C without shaking, all samples were washed twice with PBS. Thereafter, 200 mL of fresh PPGAS medium was added, and samples were sonicated to disperse the biofilm. Eradicating efficacy was assessed according to viable bacterial load determined using dilution in PBS/0.05% Tween20 and plating on agar plate to count cfu after overnight incubation at 20 °C. Three independent experiments were conducted in triplicate.

Similar MBEC assays on 24 h-old PA14 wt biofilms using 50 µg/mL of Tob either free or in combination with IA (molar [Tob]:[IA] of 1:5) were carried out for 6 h prior to the Tob diffusion studies. Three independent experiments were conducted in triplicate.

### 2.6. Tobramycin Diffusion Studies

Experiments were performed using 96-well MultiScreen Permeability Filter Plates with a polycarbonate membrane (MPC4NTR10, Sigma, Hamburg, Germany) having a pore size of 0.4 μm. Twenty-four hour-old PA14 biofilms were grown on the membrane and washed as described above prior to experiments. Inserts were placed into the companion plates, and Tob either free or in combination with IA (molar [Tob]:[IA] of 1:5) was then added to the apical compartment to reach a concentration of 50 µg/mL. The basolateral compartment was also filled with PPGAS. The solution from the basolateral compartment was collected at designated time points from 0–6 h. Tob transported through the biofilms was collected in the basolateral compartment, and its concentration was determined using LC-MS/MS. The percentage of Tob penetrating through biofilm was calculated based on the normalized 100% permeation of free Tob through the bare membrane. Three independent experiments were conducted in triplicate.

### 2.7. Tobramycin Quantification by LC-MS/MS 

The concentration of Tob was quantified using LC-MS/MS method developed in our laboratory [11]. Briefly, the system was operated by the standard software Xcalibur. To quantify Tob, a reverse phase C18 Accucore RP-MS (150 × 2.1 mm) column (Thermo Scientific, Waltham, MA, USA) was used as stationary phase. The column temperature was 30 °C. The mobile phase was composed of solvent A (water containing 0.1% *v/v* formic acid, FA) and solvent B (acetonitrile containing 0.1% *v/v* TFA), solvent ratio A:B was 95:5, and the flow rate was 300 µL/min, controlled by an Accela 1250 Pump. A standard curve was run at Tob concentrations (2.5–80 µg/mL) in PBS.

### 2.8. Statistical Analysis

All values are given as mean ± standard error of the mean (SE), from at least three independent experiments. Statistical analysis was performed with OriginPro 2017 Software (OriginLab Corp., Massachusetts, MA, USA). Significance was determined by One-way ANOVA.

## 3. Results

### 3.1. Minimum Inhibitory Concentration (MIC) Assay

The bactericidal activities of single compounds Tob, Cipro and IA, as well as their combinations Tob:IA and Cipro:IA (molar [antibiotic]:[IA] of 1:5) against PA14 wt, are shown in Table 1. The IC90 values of the combined antibiotic-IA formulations were similar to that of the corresponding free antibiotics. The IC90 values of both free Tob and Tob:IA were 3.125–6.25 µg/mL, while the obtained values for free Cipro and Cipro:IA were 3.125 µg/mL. In contrast, the use of IA alone did not show inhibitory activity against planktonic PA at the highest tested concentrations 300 µg/mL, which is also the highest IA concentration used in the further MBEC assay.

### 3.2. Pyocyanin Assay

After 16 h incubation with planktonic PA14 wt, no effects of IA on the pyocyanin production of PA14 wt were seen. The pyocyanin levels found in samples treated with IA were similar to that of PA14 wt controls (Figure 1).

### 3.3. Minimum Biofilm Eradicating Concentration (MBEC) Assay

The minimum antibiotic concentration at which the average logarithmic cfu/mL is found below the detection limit (dotted line in Figure 2), is determined as the MBEC value. The in vitro 24 h-old PA14 wt biofilms could only be completely eradicated at concentrations equal or higher than 200 µg/mL of free Tob (Figure 2A), which is remarkably higher than the MIC value against planktonic bacteria (3.125–6.25 µg/mL, Table 1). As shown in Figure 2B, the addition of IA (molar [Tob]:[IA] of 1:5) increased the efficacy of Tob against PA14 wt biofilms. When using such a combination, the MBEC value of Tob was found to be at 50 µg/mL, while the treatment with Tob concentrations 12.5–25 µg/mL exhibited significantly lower bacteria viability (cfu/mL) compared to the treatments with Tob alone. Notably, at the equivalent concentration of 12.5 µg/mL the mean log of viable cell numbers was halved from 5.10 ± 1.88 to 2.55 ± 0.72 after treatment with Tob and Tob:IA, respectively. 

The MBEC assays of Cipro and its combination with IA (molar [Cipro]:[IA] of 1:5) against PA14 wt biofilms were also performed to investigate the possible complementary effects of IA with this antibiotic. Figure 2C shows that Cipro could fully eradicate the biofilms at a concentration of 200 µg/mL. Unlike the co-administration with Tob (Figure 2B), the incorporation of IA did not enhance the efficacy of Cipro, yet showed the same eradicating effects as free Cipro (Figure 2D). Furthermore, the overall cfu/mL values at all tested concentrations in both regimens were similar. 

We performed the MBEC assays of IA against PA14 wt biofilms as controls. The highest tested concentration of 300 µg/mL was chosen, which is an equivalent amount of IA used in the combination with Tob 200 µg/mL (molar [Tob]:[IA] of 1:5). Figure 3 shows that IA alone did not have any eradicating efficacy against PA14 wt biofilms.

To investigate antibiotic permeation through bacterial biofilm, 24 h-old PA14 wt biofilms were treated for 6 h with only 50 µg/mL of Tob, i.e., well below the MBEC, either with Tob alone or in combination with IA. As can be seen in Figure 4, the incubation with Tob in both cases conditions led to a slight, but similar decrease in cfu compared to the non-treated controls. Bacterial biofilms were, however, still present under either condition as necessary for the subsequent transport studies. 

### 3.4. Tobramycin Diffusion Studies

Incomplete permeation of Tob through the 24 h-old biofilm was observed using either free or in combination with IA. The maximum amount of Tob permeating from the apical to the basolateral compartment through the 24 h-old biofilms was around 50% for free Tob (Figure 5, red line) and was reached after 3 h. However, for the combined formulation (molar [Tob]:[IA] of 1:5) the penetration of Tob was higher at all time points reaching a maximum values of nearly 80% of the initial dose. Interestingly, the maximal permeation was reached after 3 h incubation in both conditions. 

## 4. Discussion

Inhalation of a single antibiotic or combinations of systemic and nebulized antibiotics is well established in clinical practice for the treatment of PA infections in CF lungs [6]. Unfortunately, no therapeutic approach has successfully achieved the complete eradication of PA that will further prevent the recurrence of PA chronic lung infections. PA biofilms and its mucus-embedded biofilms persist and intrinsically promote resistance development [1]. Aminoglycosides, which act intracellularly, kill bacteria by binding to the 30S ribosomal sub-unit, causing a misreading of the genetic code [36]. Hence, the therapeutic efficacy, notably in the case of the polycationic Tob—the widely used first-line therapy in the treatment of PA associated infections—is restricted due to its electrostatic interactions with the negatively-charged biofilm matrix, resulting in limited and incomplete drug permeation into the bacterial biofilm core (described in Scheme 1) [37]. Furthermore, bacteria might convert into a metabolically less active phenotype in the biofilm thereby becoming less responsive to this kind of antibiotics. The results here indicate that Tob concentration of 200 µg/mL or higher are needed to fully eradicate the 24 h-old PA biofilms (Figure 2A), which is in contrast to the significantly lower MIC values (3.125–6.25 µg/mL) against planktonic form (Table 1).

Despite the short in vivo half-life, Cipro—a potent antibiotic, which inhibits bacterial DNA topoisomerase and DNA-gyrase [38]—has been formulated as a dry-powder [18] or liposomal inhalation therapy [16] for the treatment of pulmonary infections [39]. Depending on the Cipro concentrations and the biofilm growing techniques, the outcomes of the studies addressing Cipro transport through biofilms have shown heterogeneous outcomes [37,40]. To the best of our knowledge, the more conservative argument is that only a restricted quantity of Cipro could penetrate through biofilms. Consequently, the MBEC value of Cipro against the 24 h-old PA14 wt biofilm was defined at 200 µg/mL (Figure 2C), which is also dramatically higher than its MIC value of 3.125 µg/mL against planktonic bacteria (Table 1).

High nominal antibiotic dosing yet achieving poor availability inside the biofilms, probably results in sub-MIC at the site of infection, which has already been suggested to even enhance PA biofilm formation and to generate antimicrobial resistant bacterial strains [41,42,43]. With the aim of increasing the penetration of antibiotics through PA biofilms, and also to employ possible further benefits of IA in the treatment of PA infections, we investigated a novel combination of IA and Tob or Cipro with the molar ratio of [antibiotic]:[IA] = 1:5.

When conceiving this study, we first considered the chemical aspect of combining Tob and IA. We hypothesized that IA could neutralize the positive charges of Tob and thus enable a rapid and complete penetration of Tob through biofilms. The same hypothesis was applied for its combination with Cipro, in which the solubility of Cipro would especially be improved due to the excess molar ratio of carboxylic groups used in the formulation.

We found that the Tob:IA combination enhanced 4-fold the PA14 wt biofilm eradicating efficacy compared to the use of Tob alone and decreased bacterial viability in the biofilms at lower concentrations (Figure 2B). These results were encouraging and suggest that the presence of IA increases the effects of Tob, most probably by improving the accessibility of Tob into the biofilms, and/or by possibly interfering with the biological pathway of PA biofilm growth. On the complete opposite, the combination Cipro:IA did not show any improvement compared to free Cipro formulation (Figure 2D).

Notably, the highest dose of IA at 300 µg/mL did not show any activity against PA biofilms (Figure 4). Furthermore, such an IA concentration did not inhibit the growth of planktonic PA, which resulted in an OD600 of 4.23 ± 0.31 compared to 4.31 ± 0.21 of PA controls. Despite the antimicrobial properties and bacterial growth inhibition of IA [29], the normal growth of PA in the medium supplemented with IA reported here might be due to the presence of glucose in the PPGAS culture medium [30]. Within a short-term incubation, IA induces membrane stress, which PA can balance by stress-response gene up-regulation [31].

The inter-bacterial communication that takes place as a function of bacterial density via small biosynthesized molecules, the so-called *quorum* sensing (QS) process, is crucial for the establishment of complex PA biofilms [44]. Hence, using *quorum* sensing inhibitors interfering with bacterial virulence systems has shown remarkable complementary effects to antibiotics [11]. Among the virulence factors being either directly or indirectly manipulated by the *Pseudomonas* quinolone signal *quorum* sensing system, pyocyanin—a heterocyclic nitrogen-containing compound of the phenazine family—is one of the most prominent, and the evaluation of its level is well established [44]. Hence, we assessed the *quorum* sensing inhibitory effects of IA on PA14 wt using the pyocyanin assay. In the present approach, IA did not exhibit any inhibition on pyocyanin production levels of PA14 wt compared to the controls (Figure 2).

Taking together the aforementioned findings, we conclude that IA does not act alone against either in vitro planktonic or biofilm PA14 wt, and the enhancement in in vitro biofilm eradicating efficacy when complexing with Tob might be due to the other functions of IA.

Instead, it could be also possible that IA enhances the penetration of Tob through PA biofilms. To test this hypothesis, we designed a transport study through 24 h-old PA14 wt biofilms to compare the diffusion of 50 µg/mL of Tob either free or in combination with IA (molar [Tob]:[IA] of 1:5). In any transport studies, the Tob concentration was well below the previously determined 6 h MBEC. The data demonstrated the barrier function of PA biofilms, since an incomplete Tob permeation was observed in both cases compared to Tob diffusion through bare membranes (Figure 5). Notably, the antibiotic effect of Tob after 6 h of administering either free Tob or Tob:IA was not significantly different (Figure 4). However, the permeation of Tob:IA was higher than that of free Tob at all tested time points (Figure 5). Despite the relatively short study period, the steady phase in the drug transport profiles in both cases was seen after 3 h incubation. Conclusively, complexation with IA plays an important role in enhancing the biofilm penetration of Tob and thus its biofilm eradicating efficacy.

PA infections associated with CF patients also induce plenty of major clinical challenges, in particular pulmonary inflammation [45], which should be carefully considered in development of new therapeutics. The accelerating and severe inflammatory response of the CF airways is preceded at the early stage of infections. Once established, such inflammation is suggested to be associated with the bacterial burden, to impair the host immune systems [45,46]. Consequently, this worsens the lung functions and might cause severely structural airway damage [47]. Hence, the reported anti-inflammatory effects of IA could provide some additional advantage, especially under conditions of chronic PA infections [47,48] and should therefore be further investigated by more complex preclinical models.

## 5. Conclusions

The present study highlights: (i) the challenges in the treatment of bacterial infections, especially caused by major biological barrier—biofilms; (ii) and the importance of delivering drug crossing such barrier to achieve a better eradicating efficacy. Based on our data, the novel and simple combination of an antibiotic Tob with an anti-inflammatory compound IA (molar ratio [Tob]:[IA] of 1:5) enhances the PA biofilm eradicating efficacy of Tob four-fold compared to the use of Tob alone. While the same combination did not show any differences in the efficacy of Cipro. In our in vitro models, IA did not show antimicrobial or growth inhibitory effects on planktonic PA14 wt. IA neither inhibited the production of pyocyanin. We therefore conclude that the enhancement of Tob efficacy observed in complexation with IA is related to the enhancement of Tob penetration through PA biofilms. IA also showed good biocompatibility with the lung human cell line A549 (cell viability ~100% at the highest tested concentration of 1 mg/mL, data not shown) and has been reported to be safe in vivo [49]. Tob, which was FDA approved for CF treatment, is currently marketed by, e.g., Novartis Pharmaceuticals Corporation as TOBI^®^ (tobramycin inhalation solution, USP), and is administered as a nebulized solution. Technically, the addition of a small molecule such as IA is not expected to change the solution’s properties, thus not affecting the nebulization feasibility. It appears justified to further investigate this approach in some advanced disease models.

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
