# Peer review of "Itaconic Acid Increases the Efficacy of Tobramycin against Pseudomonas aeruginosa Biofilms"

_pharmaceutics, 2020, doi:10.3390/pharmaceutics12080691_

Round 1
Reviewer 1 Report
The manuscript provides evidence of a novel combination of drugs (especially tobramycin) with itaconic acid as inhibitory to P. aeruginosa biofilms, possibly by increasing biofilm penetration.
The mausccript is well written and clear and the conclusions supported by the data. I do however suggest that the authors may perform a checkerboard assay of Tob:IA to see if the interaction is truly synergistic as suggested. In addition, for the Tob diffusion studies, no statistical analyses is presented to compare the data with and without IA. I agree that the average values for normalized permeation is higher in the combination, however at all thime points there is still overlap in standard deviations.
Minor comment:
Abstract, ln 16: "raise" should be "rise"
Reviewer 2 Report
Dear Authors,
The manuscript is well written and is a very topical study in this era of antimicrobial resistance
Introduction - Not sure why nanocarriers are mentioned as the focus of the work doesn't involve their use. Can be removed and expand on other details of drug therapy: why TOBRO and CIPR, QSI, inhalation etc. The role of IA is not clear.
Methods / results - Pyocyanin why were the antibiotics and combination with IA not investiagted? Why was IA tested in MBEC assay?
In the introduction mention inhalation but no inhalation data provided and whether its dry powder inhalation proposed or nebulisation. Please provide inhalation data to demonstrate feasibility of combination therapy via inhalation
